# Premature Adult Mortality and Years of Life Lost Attributed to Long-Term Exposure to Ambient Particulate Matter Pollution and Potential for Mitigating Adverse Health Effects in Tuzla and Lukavac, Bosnia and Herzegovina

**Vlatka Matkovic [1,*], Maida Mulić [2], Selma Azabagić [2] and Marija Jevtić [3,4]**

[1]  Health and Environment Alliance, 1000 Brussels, Belgium
[2]  Public Health Institute of Tuzla Canton, University of Tuzla, 75000 Tuzla, Bosnia and Herzegovina; maida.mulic@zjztk.ba (M.M.); selma.azabagic@zjztk.ba (S.A.)
[3]  Faculty of Medicine, Institute of Public Health of Vojvodina, University of Novi Sad, 21000 Novi Sad, Serbia; marija.jevtic@uns.ac.rs
[4]  School of Public Health, Research Center for Environmental Health and Occupational Health, Université Libre de Bruxelles (ULB), 1050 Brussels, Belgium
*   Correspondence: vlatka@env-health.org

**Abstract:** Ambient air pollution is one of eight global risk factors for deaths and accounts for 38.44 all causes death rates attributable to ambient PM pollution, while in Bosnia and Herzegovina, it is 58.37. We have estimated health endpoints and possible gains if two policy scenarios were implemented and air pollution reduction achieved. Real-world health and recorded PM pollution data for 2018 were used for assessing the health impacts and possible gains. Calculations were performed with WHO AirQ+ software against two scenarios with cut-off levels at country-legal values and WHO air quality recommendations. Ambient $PM_{2.5}$ pollution is responsible for 16.20% and 22.77% of all-cause mortality among adults in Tuzla and Lukavac, respectively. Our data show that life expectancy could increase by 2.1 and 2.4 years for those cities. In the pollution hotspots, in reality, there is a wide gap in what is observed and the implementation of the legally binding air quality limit values and, thus, adverse health effects. Considerable health gains and life expectancy are possible if legal or health scenarios in polluted cities were achieved. This estimate might be useful in providing additional health burden evidence as a key component for a clean air policy and action plans.

**Keywords:** health impact assessment; mortality; life expectancy; outdoor air pollution; air pollution reduction; particulate matter; AirQ+ software; Bosnia and Herzegovina

## 1. Introduction

The world is in the global COVID-19 pandemic crisis. To date (7 September 2020), over 893,136 death outcomes have been recorded due to the coronavirus. However, despite the ever-present severity of the pandemic crisis, as well as the fact that this virus has radically changed our lives in just a few weeks, we should not forget the severity of air pollution, which will remain a serious challenge along with the societal recovery as the pandemic wave subsides.

Worldwide, the levels of air pollution translate to 3.3 million annual premature deaths, which is 5.86% of global mortality, attributable to outdoor air pollution [1]. The Global Burden of Disease Study 2017 identified ambient particulate matter (PM) pollution as one of eight global risk factors

for deaths, with 38.44 (32.75–43.93) per 100,000 population all causes deaths attributable to ambient PM pollution [2].

The World Health Origination (WHO) estimates the death rate due to ambient air pollution in the European region at 48.10 (56.45–40.31). In Bosnia and Herzegovina (BiH), ambient particulate matter pollution is a leading environmental risk factor with 58.37 (35.10–82.38) all causes deaths attributable to PM pollution. It is a considerable change of +151.63% in the mortality rate due to air pollution from reported data in 1991 (23.2 (14.34–34.51) per 100,000 population). In 2018, life expectancy at birth was 72.94 years in Tuzla. Lukavac city has a higher life expectancy at birth, 74.94 years. While life expectancy in BiH in 2018 was 77.26.

In the complex political division of Bosnia and Herzegovina, there are two entities: the Federation of BiH (FBiH) and Republika Srpska. The two entities run separate air monitoring systems as well as health systems. The area of the Tuzla Canton is characterized by one of the largest coal basins in BiH. In the past 50 years, Tuzla has been home to the country's largest coal power plant, which is located half-way between the centers of Tuzla (8 kilometers) and Lukavac (7 kilometers). The plant fed several ash and slag disposal sites. At two of the ash disposal sites, remediation was performed in a substandard way. There has been strong civil opposition to the opening of the new ash disposal site, demanding the closure of the existing ones in a proper manner [3].

The review of the evidence on health aspects of air pollution by the WHO concludes that any reduction in $PM_{2.5}$ concentration will result in public health benefits, whether or not the current levels are above or below the limit values [4]. The WHO guidelines suggest achieving a $PM_{2.5}$ annual mean of 10 µg m$^{-3}$ for significant reductions in risks for acute and chronic health effects. The WHO guideline values for air pollution are being revised at the time of writing this paper and may suggest even lower levels of air pollution exposure to protect human health [5]. New research in low-exposure environments confirms no apparent threshold below which the effects cease to exist [6,7].

The legal framework on air quality in the FBiH [8] has air quality legislation aligned with the EU Air Quality Directive. The legislation sets yearly limit values for particulate matter, both $PM_{10}$ (40 µg m$^{-3}$) and $PM_{2.5}$ (25 µg m$^{-3}$). It also regulates $PM_{10}$ pollution with a 24-h limit of –50 µg m$^{-3}$, not to be exceeded more than 35 days per year. However, legal limits are not being achieved across the country and over the years, as outlined in the Hydrometeorological Institute of FBiH reports on air quality. Public policy on ambient pollution, when implemented, and its ambitions on the limit values can determine the magnitude of the mitigation capacity of the adverse health effects and mortality among the populations.

This study aimed to estimate the impact of ambient PM pollution on the mortality, life expectancy (LE), years of life lost (YLL) in two cities in BiH (Tuzla and Lukavac). Furthermore, we estimated possible gains in LE and YLL that would result from the reduction in ambient PM pollution by two scenarios. The first scenario is to comply with the national air quality legislation, while in the second scenario, the WHO ambient PM pollution guidelines will be achieved.

## 2. Methodology

### 2.1. Study Area and Population

In this study, we estimated health endpoints and health gains in two air pollution reduction scenarios for two municipalities in the Tuzla Canton, Tuzla and Lukavac, for which sufficient air quality data were available.

Tuzla is the third-largest city in BiH, covering an area of 294 km$^2$. It is the administrative center of the Tuzla Canton of the FBiH. According to the latest available data from the official Census in 2013, it had 110,979 inhabitants. The city's population has 66.9% of adults aged 30 or above, while 14.7% are children between 5–19 years of age.

Lukavac is the fourth biggest municipality of the Canton with a population of 44,520, of which 65.6% of adults aged 30 or above, while 16.0% are children between 5–19 years of age.

## 2.2. Ambient Particulate Matter Pollution Data

The FBiH entity has a total of 21 monitoring stations that are run and managed by the Federal Hydrometeorological Institute. Based on the 2018 report of the Hydrometeorological Institute, the average $PM_{2.5}$ pollution for FBiH for 2018 was 38.8 µg m$^{-3}$.

We looked specifically for pollution and health data from the Tuzla Canton. In the Tuzla Canton, three out of five stations were eligible to include the data: two stations in Tuzla (Bukinje and Skver) and one station in Lukavac. The Bukinje industrial station is located at latitude 44°31′26″ N and longitude 18°36′01″ E at a 214 m altitude. The Skver station is located at latitude 44°33′28″ N and longitude 18°40′25″ E at a 234 m altitude. The BKC station (latitude 44°31′56″ N and longitude 18°39′18″ E at a 231 m altitude) did not provide data for 2018 and was, thus, excluded from the analysis. Lukavac has one air quality measuring station, an industrial station, located at latitude 44°32′00″ N and longitude 18°32′05″ E at a 187 m altitude. One more station was available in the Canton, Zivnice, but due to it only providing 69.7% of the data for 2018, we excluded it from the analysis, as per the Aphekom project guidelines for calculating health impacts and availability of data [9].

In this analysis, we obtained the air pollution data from the Federal Hydrometeorological Institute hourly measured $PM_{2.5}$ dataset for the stations Bukinje, Skver, and Lukavac. Air pollution datasets are available through the request for access to information from the Federal Hydrometeorological Institute. This air pollution data set represents 95.8% of the total yearly coverage of maximum possible 8760 h measured in a year. We excluded all missing (*n* = 1356; 56%) and negative (*n* = 995; 39%) values from the hourly dataset. In BiH, limit values for $PM_{2.5}$ were 25 µg m$^{-3}$ for $PM_{10}$ 40 µg m$^{-3}$ with the 24-h limit at 50 µg m$^{-3}$ not to be exceeded more than 35 times in a calendar year. Our dataset for 2018 showed there were exceedances of $PM_{10}$ equivalent (using the WHO country conversion factor for Bosnia and Herzegovina 0.76) out of the total of 126 days in Tuzla and 170 days in Lukavac.

In 2018, the $PM_{2.5}$ annual mean for Tuzla was 39.36 ± SD 36.48 (SD, standard deviation) µg m$^{-3}$, and for Lukavac 52.96 ± SD 47.04 µg m$^{-3}$ (Table 1). The $PM_{2.5}$ concentration in Tuzla ranged from 2.69 to 222.97 (min–max), and in Lukavac, the range was from 4.49 to 312.28 (min–max).

**Table 1.** Annual mean concentration, cut-off value for legal scenario and health-protective scenario.

| Pollutant | Tuzla Annual Mean | Lukavac Annual Mean | Legal Scenario Cut-Off Value, Legal Limits (BiH and EU) | Health-Protective Scenario Cut-Off Value, WHO Guidelines |
|---|---|---|---|---|
| $PM_{2.5}$ (in µg m$^{-3}$) | 39.38 | 52.94 | 25 | 10 |

We based health impacts assessments, in terms of LE and YLL, on concentrations presented in Table 1.

## 2.3. Mortality Data in Tuzla Region

The source of mortality data and baseline incidence (BI) expressed as the rate per 100,000 inhabitants (shown in Table 2.) and the Life Tables for 2018 for all the cities in the Tuzla region was provided by the Tuzla Canton Public Health Institute on request for access to the information procedure [10]. In the Life Tables, a total of 18 age ranges were divided into 5-year intervals from age ≤4 years to age ≥85 years.

These data were used to estimate attributable mortality, LE, and YLL due to ambient PM pollution. Estimates of the life expectancy (LE) and years of life lost (YLL) and possible gains were the results of the impact of the levels of $PM_{2.5}$ on population mortality. YLL measures the years lost through premature mortality and is the component of the Disability Adjusted Life Years (DALY), a measure of overall disease burden that represents an indicator of life expectancy. Mortality in Tuzla was the highest in the region (Canton), with a 1220.95 death rate per 100,000 population–equal to 1355 cases in 2018. Tuzla was followed by Lukavac, where the death rate was 1096.14 per 100,000 population. The average death rate in the Tuzla region was 985.11 per 100,000 population.

**Table 2.** Mortality cases and baseline incidence (BI) by all municipalities of the Tuzla region in 2018.

| Municipalities in Tuzla Region | Number of Deaths in 2018 | BI |
|---|---|---|
| Banovići | 189 | 829.93 |
| Čelić | 112 | 1066.46 |
| Doboj Istok | 106 | 1034.35 |
| Gračanica | 408 | 902.26 |
| Gradačac | 393 | 998.98 |
| Kalesija | 238 | 720.06 |
| Kladanj | 120 | 971.82 |
| Lukavac * | 488 | 1096.14 |
| Sapna | 76 | 679.91 |
| Srebrenik | 343 | 864.46 |
| Teočak | 73 | 983.3 |
| Tuzla * | 1355 | 1220.95 |
| Živinice | 483 | 836.15 |
| Total number of deaths | 4384 | 985.11 |

BI is a crude rate per 100,000 inhabitants, * Municipalities with air quality monitoring stations.

### 2.4. AirQ+ Calculation against Mitigation Scenarios

In this study, we used AirQ+ software version 2.0 to estimate the mortality, life expectancy (LE), and years of life lost (YLL) as a result of exposure to $PM_{2.5}$. The WHO AirQ+ tool has been developed by the WHO Regional Office for Europe, European Centre for Environment and Health (ECEH), Bonn office, Germany to calculate the magnitude of the burden and impacts of air pollution on health in the population. All calculations performed by AirQ+ are based on methodologies and concentration–response functions established by epidemiological studies.

Steps of the calculation included input data on: (1) the air quality dataset as daily mean concentrations for every day in 2018; (2) exposed population; (3) number of cases for total mortality; (4) population at risk, for mortality aged 30+; (5) for calculating life expectancy and years of life lost, were inputted into Life Tables for 2018. In this study, we did not control for other variables, such as socioeconomic status, risk behaviors, that might affect people's health and life span.

Afterward, we estimated the mortality based on pollutant-health outcome pairs concentration-response functions by WHO for $PM_{2.5}$ Mortality, all (natural) causes (adults age 30+ years) Relative Risk (RR) and 95% Confidence Interval (CI) applied 1.062 (95% CI 1.04–1.083) with log-linear shape and β coefficient 0.006015392 (lower–upper, 0.003922071–0.007973497) per 10 μg m$^{-3}$ increase of $PM_{2.5}$. The concentration-response functions used in the software were based on the systematic review of all studies available until 2013 and their meta-analysis [11].

The health effects were calculated against two scenarios with cut-off levels presented in Table 1. First, the legal limits scenario: levels of PM pollution would meet the legal limits in BiH and the EU, set at 25 μg m$^{-3}$ for $PM_{2.5}$; less ambitious scenario. Second, health-protective scenario: where the level of pollution would not exceed the recommendations of WHO, set at 10 μg m$^{-3}$ for $PM_{2.5}$, the ambitious scenario.

### 3. Results

Table 3 shows the attributable proportion and numbers of mortality cases due to PM pollution in Tuzla and Lukavac. The mortality due to ambient PM pollution was calculated against two scenarios, legal and health-protective. If we look at the less ambitious scenario, legal, ambient $PM_{2.5}$ pollution was responsible for at least 8.29% and 15.47% of all-cause mortality among adults in Tuzla and Lukavac, respectively. Attributable cases per 100,000 population are 101 for Tuzla and 169 for Lukavac in the legal scenario. In the health-protective scenario, the attributable proportion and the number of mortality cases would be considerably higher.

**Table 3.** Mortality, all (natural) causes (adults age 30+ years) attributable to $PM_{2.5}$ pollution in Tuzla and Lukavac, in legal and health-protective scenarios.

| Measure | Tuzla | | | | Lukavac | | | |
|---|---|---|---|---|---|---|---|---|
| | Legal Scenario, $PM_{2.5}$ Cut-Off Value 25 µg m$^{-3}$ | | Health-Protective Scenario, $PM_{2.5}$ Cut-Off Value 10 µg m$^{-3}$ | | Legal Scenario, $PM_{2.5}$ Cut-Off Value 25 µg m$^{-3}$ | | Health-Protective Scenario, $PM_{2.5}$ Cut-Off Value 10 µg m$^{-3}$ | |
| | Central Value | (Uncertainty Range) | Central Value | (Uncertainty Range) | Central Value | (Uncertainty Range) | Central Value | (Uncertainty Range) |
| Attributable Proportion | 8.29% | (5.49–10.84%) | 16.20% | (10.89–20.89%) | 15.47% | (10.38–19.97%) | 22.77% | (15.5–29.0%) |
| Number of Attributable Cases | 75 | (50–98) | 147 | (99–189) | 50 | (33–64) | 73 | (50–93) |
| Number of Attributable Cases per 100,000 Population at Risk | 101 | (66–132) | 197 | (132–255) | 169 | (113–218) | 249 | (169–317) |

Tables 4 and 5 present years of life lost (YLL) of the population in Tuzla and Lukavac attributed to the long-term exposure to $PM_{2.5}$ of 39.38 µg m$^{-3}$ and 52.94 µg m$^{-3}$, respectively. YLL was calculated for both locations against two scenarios with cut-off values of 25 µg m$^{-3}$ and 10 µg m$^{-3}$ for $PM_{2.5}$. In 2018, 105.63 (95% CI 70.82–136.43) were lost in Tuzla for all ages. Over 10 years, if health-protective air pollution levels of 10 µg m$^{-3}$ were achieved, 9148.16 YLL for all ages could be prevented. Over the same period, if PM pollution complied with legal limits, we would see 4600.12 YLL prevented.

**Table 4.** Years of life lost (YLL) due to premature mortality for Tuzla, against two cut-off scenarios.

| Years of Life Lost (Ages) | Tuzla | | | |
|---|---|---|---|---|
| | Legal Scenario, $PM_{2.5}$ Cut-Off Value 25 µg m$^{-3}$ | | Health-Protective Scenario, $PM_{2.5}$ Cut-Off Value 10 µg m$^{-3}$ | |
| | Central Value | (Uncertainty Range) | Central Value | (Uncertainty Range) |
| YLL for year 2018 (all ages) | 53.87 | (35.61–70.49) | 105.63 | (70.82–136.43) |
| YLL for year 2019 (all ages) | 158.76 | (104.87–207.9) | 311.95 | (208.85–403.42) |
| YLL over 10 Years-2028 (all ages) | 4600.12 | (3026.14–6046.8) | 9148.16 | (6074.95–11,918.88) |
| YLL over 20 Years-2038 (all ages) | 15,456.08 | (10,138.26–20,372.04) | 31,006.47 | (20,467.97–40,624.13) |

For Lukavac in 2018, 53.08 YLL were lost due to PM pollution. When pollution cut would reach WHO guidelines over a period of 10 years, there would be a prevention potential of 4355.16 YLL, while under the legal scenario, 2906.96 YLL could be avoided.

The life expectancy (LE) at birth at current (2018) levels of $PM_{2.5}$ pollution in Tuzla and Lukavac is presented in Table 6. We also present life expectancy at birth if the legal values of 25 µg m$^{-3}$ and WHO guidelines of 10 µg m$^{-3}$ were achieved.

**Table 5.** Years of life lost (YLL) due to premature mortality for Lukavac, against two cut-off scenarios.

| Years of Life Lost (Ages) | Lukavac | | | |
|---|---|---|---|---|
| | Legal Scenario, $PM_{2.5}$ Cut-Off Value 25 µg m$^{-3}$ | | Health-Protective Scenario, $PM_{2.5}$ Cut-Off Value 10 µg m$^{-3}$ | |
| | Central Value | (Uncertainty Range) | Central Value | (Uncertainty Range) |
| YLL for year 2018 (all ages) | 35.93 | (24.04–46.5) | 53.08 | (36–67.85) |
| YLL for year 2019 (all ages) | 104.75 | (69.96–135.77) | 155.14 | (104.94–198.72) |
| YLL over 10 Years-2028 (all ages) | 2906.96 | (1926.89–3794.26) | 4355.16 | (2912.42–5637.82) |
| YLL over 20 Years-2038 (all ages) | 9779.17 | (6456.69–12,812.19) | 14,742.71 | (9797.76–19,199.59) |

**Table 6.** Life expectancy (LE) at birth at current $PM_{2.5}$ pollution levels and LE against legal and health-protective scenarios.

| | Tuzla | | | | Lukavac | | | |
|---|---|---|---|---|---|---|---|---|
| | Legal Scenario, $PM_{2.5}$ Cut-Off Value 25 $\mu g\ m^{-3}$ | | Health-Protective Scenario, $PM_{2.5}$ Cut-Off Value 10 $\mu g\ m^{-3}$ | | Legal Scenario, $PM_{2.5}$ Cut-Off Value 25 $\mu g\ m^{-3}$ | | Health-Protective Scenario, $PM_{2.5}$ Cut-Off Value 10 $\mu g\ m^{-3}$ | |
| Life Expectancy at Birth | Central Value | (Uncertainty Range) | Central Value | (Uncertainty Range) | Central Value | (Uncertainty Range) | Central Value | (Uncertainty Range) |
| With current air pollution | 72.94 | | | | 74.94 | | | |
| If scenario were achieved | 73.97 | (73.61–74.30) | 75.03 | (74.31–75.70) | 76.53 | (75.98–77.03) | 77.36 | (76.53–78.13) |
| Gain in years if scenario were achieved | 1.0 | (0.67–1.36) | 2.1 | (1.37–2.76) | 1.6 | (1.04–2.09) | 2.4 | (1.59–3.19) |

There would be a gain of one year (lower–upper, 0.67–1.36) of life (73.97 years) if pollution were at legal limits. We would see a greater gain in years when $PM_{2.5}$ pollution is cut down to WHO guidelines of 10 $\mu g\ m^{-3}$. Life expectancy in a health-protective scenario would be 75.03 years, or 2.1 (1.37–2.76) years would be gained.

If legal air pollution levels of 25 $\mu g\ m^{-3}$ were achieved, life expectancy would increase by 1.6 (1.04–2.09) year, while in the health-protective scenario, 2.4 (1.59–3.19) years more would be gained.

To complement results on LE at birth, Table 7 explains expected life remaining (ELR) and delta ELR, which is interpreted as a gain in years when pollution is cut down to legal and health-protective scenarios, 25 $\mu g\ m^{-3}$ and 10 $\mu g\ m^{-3}$, respectively. A person of 30 years of age in Tuzla is expected to live another 45.64 years. If the legal limits of air pollution were achieved, this person would gain 0.84 years, and under the health-protective scenario, 1.71 years of life would be added to life expectancy. In Lukavac, a 30-year-old is expected to live slightly less, 46.59 years more. One point three-six years would be gained with legal PM limits, and 2.09 years if health-protective levels were achieved.

**Table 7.** Expected life remaining (ELR) and gain, in years, under the legal and health-protective scenario for Tuzla and Lukavac.

| | Tuzla | | | Lukavac | | |
|---|---|---|---|---|---|---|
| Age | ELR (Years) | Gain in Life Years, Legal Scenario | Gain in Life Years, Health-Protective Scenario | ELR (Years) | Gain in Life Years, Legal Scenario | Gain in Life Years, Health-Protective Scenario |
| 0 | 72.94 | 1.03 | 2.09 | 74.94 | 1.59 | 2.42 |
| 1 | 72.37 | 1 | 2.03 | 74.09 | 1.56 | 2.39 |
| 30 | 45.64 | 0.84 | 1.71 | 46.59 | 1.36 | 2.09 |
| 35 | 41.01 | 0.81 | 1.66 | 41.67 | 1.35 | 2.07 |
| 40 | 36.41 | 0.79 | 1.61 | 36.73 | 1.34 | 2.06 |
| 45 | 31.86 | 0.76 | 1.56 | 32.31 | 1.27 | 1.96 |
| 50 | 27.64 | 0.71 | 1.47 | 27.86 | 1.21 | 1.86 |
| 55 | 23.52 | 0.66 | 1.36 | 23.8 | 1.1 | 1.69 |
| 60 | 18.53 | 0.66 | 1.36 | 18.83 | 1.1 | 1.69 |
| 65 | 14.82 | 0.6 | 1.24 | 15.07 | 0.97 | 1.5 |
| 70 | 12.01 | 0.5 | 1.04 | 11.59 | 0.82 | 1.27 |
| 75 | 8.71 | 0.43 | 0.89 | 8.09 | 0.68 | 1.06 |
| 80 | 5.59 | 0.37 | 0.77 | 4.94 | 0.54 | 0.85 |
| 85 | 3.67 | 0.31 | 0.65 | 2.3 | 0.42 | 0.67 |

## 4. Discussion

In this study, we estimated mortality, life expectancy, and years of life lost attributed to long-term ambient $PM_{2.5}$ pollution in two cities in BiH, Tuzla and Lukavac. We also looked at the LE and possible gains in LE if current air quality legislation were implemented. This means that air pollution would be reduced to the legal limits. The second scenario includes PM pollution reduction to the WHO guidelines values. Our baseline data was based on real-world recorded health and PM pollution data for 2018.

We estimated that 16.20% and 22.77% of premature deaths were due to ambient PM pollution in Tuzla and Lukavac, respectively. This translates to 197 and 249 premature deaths for the respective cities. The higher attributable proportion for mortality in Lukavac compared to Tuzla stems from the fact that the citizens in Lukavac are exposed to higher concentrations of $PM_{2.5}$ pollution. Hence, life expectancy gains would be greater if the air pollution decrease from 52.94 μg m$^{-3}$ to healthy levels (10 μg m$^{-3}$) in comparison to Tuzla, where air pollution decreased from 39.38 μg m$^{-3}$ to healthy levels.

These death rates were much larger than the ones calculated by EEA in their annual Air Quality Report [12]. EEA estimated that for BiH, the annual $PM_{2.5}$ mean in 2018 was 18.9 μg m$^{-3}$, which results in premature death rates of 97 per 100,000 population for BiH. The value is expressed as the population-weighted concentration, not only from monitoring stations but also including the chemical transport model results and other supplementary data according to the methodology described by The European Topic Centre on Air Pollution and Climate Change Mitigation Consortium [13]. It has been reported that this methodology tends to underestimate high values, and additional exceedances could, therefore, be expected in countries such as BiH (as a relatively large fraction of the population lives in areas with concentration levels above 30 μg m$^{-3}$) [14]. The Global Burden of Disease (GBD) study showed a smaller health burden than our study. The GBD presented that 58.3 per 100,000 population all-cause death rate in 2017 was attributable to ambient PM pollution in BiH [15].

The main goal of the legally imposed levels of maximum concentrations of PM levels is to prevent adverse health effects on the population and environmental protection. In reality, there is a wide gap in what is observed and the implementation of the legally binding air quality limit values. The gap is even wider on reaching air quality values to protect human health proposed by WHO. Due to PM pollution in Tuzla and Lukavac, LE was lower by 1 and 1.6 years, respectively, than it could be improved if legal air quality limits were achieved in the region. The health-protective or WHO guidelines brought an even longer life expectancy for the two cities, 2.1 and 2.4 years, respectively. Furthermore, life expectancy in BiH has been slowly rising since the 1960s, with the exception of the years of war in the country in the 1990s. Our data show that this increase in life expectancy of the Tuzla and Lukavac population could be offset by 2.1 and 2.4 years, respectively, for the two cities as a result of the burden from ambient PM pollution. Similarly, the Aphesis study suggested that exposure even to low doses of PM over long periods reduces life expectancy.

Public health data and health burden evidence is becoming a key component in policy decisions and action plans, as presented in papers for Greece, Italy, and Iran [16–19] that can clearly bring considerable health gains for the population. However, contributions to air pollution by sectors needs to be examined, and data made publicly available to steer policy recommendations and decisions.

In the case of BiH, the authorities have failed to determine the exact contribution from each sector, and BiH is one of the few countries that does not report their emissions contributions to the Convention on Long-Range Transboundary Air Pollution registry, except for the year 2018. In that report for 2018, Bosnia and Herzegovina disclosed air emission contributions [20]. In Tuzla, the main dust emitter was the coal power plant, with 776 tons/year of emissions of dust to the air. In Lukavac, several emitters of dust are contributing to the air pollution: the manufacturer of ammonia soda (160 t/y), the coke manufacturing, fertilizer, and water treatment facility (55 t/y), and cement production (27 t/y). Belis et al. estimated major pollution sources in the Western Balkans region [21]. In their model, they reported that energy production in inefficient coal-fueled power plants (22%) was identified as

one of the main sources of $PM_{2.5}$ in the Western Balkans, followed by agriculture (19%), residential combustion (16%), and road transport (7%).

Similarly, different groups have assumed that the main causes of ambient air pollution are the local Tuzla thermal power plant with four units, followed by individual household heating and traffic [22]. Cutting pollution from the main sources would most likely bring sizeable improvements in air quality and, thus, in the health status of the local population. BiH ratified the Paris Agreement in March 2017. The country is still drafting the Nationally Determined Contributions plan and has yet to begin implementing the Paris Agreement systematically [23]. Within the Energy Community Treaty, BiH committed to the emission reduction in the energy sector for different pollutants and to close down the Tuzla plant (blocks 3 and 4) by the end of 2023 [24].

The overestimation of health impacts might be due to high baseline mortality rates. However, when compared to EUROSTAT [25] data on the mortality rates, it showed that BI mortality in our study was above the European average of 1050 per 100,000 population for the year 2018, but lower than some of the neighboring countries; Croatia 1290; Serbia 1460 per 100,000 population for the year 2018. The slightly higher BI mortality could be driven by external causes, such as tobacco. The WHO estimated that in 2018 in BiH, the prevalence of tobacco smoking in persons age 15 or older in BiH was 33.6%, while the European average was 28.7% [26]. On the other hand, the toxicity of the PM pollution in this region may be different from those exposed to other cohorts in the world on which exposure-response functions are based [27–29].

There are a number of uncertainties in estimating the benefits of cutting the exposure to (PM) pollution. Some of the uncertainties are described more in-depth by different authors in previous works related to estimations of health outcomes of air pollutants [30,31]. Consideration needs to be given to the uncertainties in the exposure-response functions used to link annual average $PM_{2.5}$ with a percent change per $\mu g\ m^{-3}$ in mortality hazard rates for different cohorts and as the science advances [7,32]. Here used exposure-response function was based on epidemiological studies for long-term $PM_{2.5}$ exposure carried out in locations and populations other than those considered here. However, we used those best described by Héroux and widely used in the European context [11]. The use of a risk factor of 6% in this study was, most likely, not too high, which comes from cohort studies reporting higher coefficients, 13–17% increase in mortality hazards, per 10 $\mu g\ m^{-3}$ $PM_{2.5}$ [27]. Important uncertainty lies in the transferability of the estimates between the Relative Risk referring to the European population to the Relative Risk that would be best suited for the context of Bosnia and Herzegovina. The toxicity of $PM_{2.5}$, populations and its characteristics (smoking status, socioeconomic conditions, diet, time spent outdoors, energy poverty, and housing characteristics), weather, access to health care, and other factors may differ between these two contexts. Moreover, poorly planned services and weak governance in the country negatively impacts quality and outcomes of health care [33] and, thus, may contribute to varied relative risk than is suggested for the European population. However, substantial bias is unlikely as the original evidence is based on studies of a mix of heterogeneous observations.

Furthermore, our $PM_{2.5}$ pollution dataset from three monitoring stations indicates exposure in the two cities. Factors likely to affect individual exposures, such as personal time-activity patterns, were not taken into account by these data. This number of monitoring stations might not suffice to represent the general $PM_{2.5}$ situation in this and similar regions. Studies on the variation of urban pollution showed that the spatial and temporal variation of the $PM_{2.5}$ concentrations is moderate and is mostly of regionally and long-range transported origin [34].

The benefits of reducing air pollution might not directly lead to health benefits over time as we have estimated the long-term effects of air pollution on the population of Tuzla and Lukavac, assuming that the air pollution levels did not change. However, interventional studies in Dublin and Hong Kong showed a reduction in mortality in years immediately following the reduction in pollution [35,36].

While air pollution's contribution to mortality is one of the crucial datasets in effective policies that will benefit public health, useful information should be obtained on the health burden in terms of morbidity and excess burden to the local health care centers. Health centers and the health professionals

providing services in them are under pressure and often lack the capacity and the means to treat patients. Reductions in the need for health care services and, thus, pressure on the local health care systems could lead to better performances and preparedness to answer other health demands, as we see today with COVID-19. Future research shall explore the actual link of morbidity due to air pollution and the health care service use. Air pollution reduction could be one of the apparent prevention policies advocated by the health community with clearer real-world links to health care systems burden of air pollution mortality and morbidity.

**Author Contributions:** Conceptualization, V.M.; methodology, V.M.; validation, S.A., M.J.; formal analysis, V.M.; resources, V.M., M.M. and S.A.; data curation, M.M., S.A. and V.M.; writing—original draft preparation, V.M.; writing—review and editing, V.M., M.M., S.A., M.J.; supervision, V.M. All authors have read and agreed to the published version of the manuscript.

**Funding:** This research received no external funding.

**Acknowledgments:** We kindly acknowledge Denis Žiško (Center for Ecology and Energy, Tuzla, Bosnia and Herzegovina) and Srđan Kukolj (Health and Environment Alliance) for their support in collecting the data.

**Conflicts of Interest:** The authors declare no conflict of interest.

## Abbreviations

Bosnia and Herzegovina (BiH), disability-adjusted life years (DALYs), expected life remaining (ELR), Federation of Bosnia and Herzegovina (FBiH), life expectancy (LE), particulate matter (PM), World Health Origination (WHO), years of life lost (YLL)

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
