# Peer review of "Premature Adult Mortality and Years of Life Lost Attributed to Long-Term Exposure to Ambient Particulate Matter Pollution and Potential for Mitigating Adverse Health Effects in Tuzla and Lukavac, Bosnia and Herzegovina"

_atmosphere, doi:10.3390/atmos11101107_

Round 1

Reviewer 1 Report

General comments This paper evaluates the impact of air pollution on people's health in Tuzla and Lukavac, Bosnia, and Herzegovina areas. It estimates how the health and life expectancy of people in these areas will be improved if effective air control policies are implemented. The topic of this article is important and it should provide useful ideas for the formulation of air control policy in Tuzla and Lukavac. However, many expressions in this paper are not clear and the results need a more profound analysis. In addition, there are a lot of grammar and spelling problems in the article, the authors need to rewrite some parts of the article. Major comments 1. The authors need to clarify the sources of the pollutant concentration and mortality data used in this paper, any reference? 2. It is necessary to describe the specific input and output of AirQ+ software and the particular steps of calculation. In the process of analysis, how to control other variables that might affect people's health and life span? 3.This paper lists the attributions of mortality (Table 3) under the 10 μg/m3 scenario. I suggest the authors to compare the results with 25 μg/m3 and current scenarios, which should be useful for the policy makers. 4.Please provide some potential reasons behind the differences in Attributable Proportion, YLL, and LE between Tuzla and Lukavac. For example, Lukavac has higher pollution concentration and higher attributable mortality. However, why does YYL for Lukavac turns to be worse than that for Tuzla under the 10 μg/m3 scenarios? 5.No figure is presented in this paper. It would be better if the authors can present some of the results, such as those in Tables 4 and 5, in figures. 6. In the discussion section, the author's statement deteriorates the credibility of the results in this paper, such as Line 241-264. In addition, it is better to describe Line 196-206 in the introduction section. Specific comments 1. Commas A comma should be added after the following places in the article: Line 22 “In the pollution hotspots“ Line 41 "In Bosnia and Herzegovina (BiH)" Line 71 "while in the second scenario" Line 88 "Based on the 2018 report of the Hydrometeorological institute" Line 90 "In Tuzla canton" Line 151 “Over the period of 10 year“ “if health protective air pollution levels would be achieved of 10 μg/m3” Line 155 “For Lukavac in 2018” Line 155 “When pollution cut would reach WHO guidelines over period of 10 year” Line 157 "while under the legal scenario" Line 165 "In Tuzla" "with current levels of PM pollution of 39.38 μg/m3" Line 172 “while in a health protective scenario” Line 178 "Table 7 explains Expected Life Remaining (ELR) and delta ELR" Line 179 "25 μg/m3 and 10 μg/m3" Line 180 "When legal limits of air pollution would be achieved" Line 181 "and under health protective scenario" Line 196 “EEA estimated that for Bosnia and Herzegovina” Line 197 “annual PM2.5 mean in 2018 was 18.9 μg/m3” Line 225 "Public health data and health burden evidence is becoming a key component in policy decisions and action plans" Line 229 "the authorities have failed to determine the exact contribution from each sector" Line 236 "on the mortality rates" Line 251 "on the population of Tuzla and Lukavac" 2. Singular or plural Line 14 "Ambient air pollution is eight global risk factors"  "Ambient air pollution is one of eight global risk factors" Line 26 "are key component"  "is a key component" Line 74 "represent"  "represents" Line 109 "concentrations"  "concentration" Line 135 "Health effect were"  "Health effects were" Line 144 "Attributable cases per 100,000 population is 191"  "Attributable cases per 100,000 population are 191" Line 150 "YLL were calculated"  "YLL was calculated" Line 150 "Both location"  "Both locations" Line 151 "Over the period of 10 year"  "Over 10 years" Line 181,182 "1.71 year" “46.59 year" "1.36 year"  “years” Line 212 "The main goal of ... are"  "The main goal of ... is" Line 21, 220 "Our data shows"  "Our data show" Line 222 "exposure even to ... reduce"  "exposure even to ... reduces" Line 244 "There are number of"  "There is a number of" 3. Definite article There are many missing definite articles in the paper. The following lists some sentences that need to be added with definite articles, but they are not complete. Please check the other parts of the manuscript carefully. Line 19 "with a cut of" Line 32 "due to the coronavirus" Line 34 "forget the severity of air pollution" Line 40 "In the World Health Origination(WHO) ... the death rate ..." Line 42 "is leading a environmental risk factor" Line 43 "It is a considerate change of " Line 51 "in a substandard way" Line 71 "The first scenario" Line 87 "Entity FBiH had a total of ... by the Federal Hydrometeorological Institute." 4. Symbol Please pay more attention to the use of superscript and subscript, such as PM2.5 and μg/m3. Please check the correct format of symbols in the paper. 5. Line 190 "Thus" cannot be used as a conjunction between two sentences. “ ... would be implemented; thus,” or “ ... would be implemented. Thus," 6. Line 177 "To compliment results" "To complement results" 7. The below sentences need to be rewritten to make the structure and expression clearer. Line 68-71 "The aim of this study was to estimate the attributable proportion of mortality to ambient PM pollution in two cities in BiH (Tuzla and Lukavac); and to assess Life Expectancy (LE) and Years of Life Lost (YLL) and possible gains that would result with the reduction of ambient PM pollution by 2 scenarios." Line 101-102 "The data set represents 95.8% of total measurement coverage." Line 103-105 "Legal framework of BiH sets annual mean concertation for PM2.5 at 25 μg/m3, for PM10 at 40 μg/m3 with the 24-hour limit at 50 μg/m3 not to be exceeded more than 35 times in a calendar year. " Line 109-110 "Minimum PM2.5 concentrations in Tuzla was 2.69 and maximum 222.97, while in Lukavac min was 4.49 and max was 312.28. " Line 209-211 "Air pollution is eight risk factor for the country that drive the most death and disability combined and it was estimated by GBD to be in past 10 years (1007-2017) reduced by 10.4%." 8. Line 55-58 It is better to add references appropriately. 9. Line 61-65 Please add the references. 10. Line 72-74 Perhaps it is better to explain YLL and DALY in the methodology section. 11. Line 90 “Tuzla canton”  "Tuzla Canton". Please revise the similar issue throughout the manuscript.

Reviewer 2 Report

This paper presents health impacts analysis for in Tuzla and Lukavac of Bosnia and Herzegovina, due to particulate matter. Paper covers the baseline information necessary calculating these impacts and results for current and two future (ideal) scenarios.

The methodology is clearly explained, authors show a comparison with global assessments (GBD), and describe the range of these impacts on various age groups.

The uncertainties in estimating health impacts are also explained clearly in the discussion section.

Minor comments:

1. what is the need to use 2-decimal points when presenting the results, given the level of uncertainties at all the levels - from monitoring data, to baseline mortality, to IER's in the AirQ+ model

2. The lives and life years saved by complying to EU and WHO norms is a good marker for the analysis. While there is no consensus on the source contributions from the officials or the academic institutions, it will be good to have a table on what is available in this regards. Given some information is available on the source contributions from regional and global models, what is the level of commitment required in various sectors to reach these EU and WHO norms.

Round 2

Reviewer 1 Report

I am generally satisfied with the revision but I still have two concerns about the method/data used in this work:

1) According to Line 120, the PM2.5 data were only from three stations. As we know, the PM2.5 concentration is influenced by emission, terrain and meteorological condition. The data from three stations cannot represent the general PM2.5 situation in these regions. Uncertainty discussion is needed. 

2) Lines 169-174, the concentration-response function in the software is for European Union based on the reference cited by the authors. Since this function is influenced by the medical-care situation, please discuss whether this function is suitable for the regions analyzed in this work. 

Minor comments:

1) Add lines to the bottom of Tables 4 and 5.

2) Please merge Lines 263-265 to nearby paragraph.

3) Please enlarge the font size of the table caption.
